# Differences in soil carbon fractions and microbial communities and their underlying mechanisms between assisted natural regeneration and plantation forests in subtropical China

Zhiqiang Ding[1,2], Zhijie Yang[1,2], Yusheng Yang[1,2]*

**1** School of Geographical Sciences, Fujian Normal University, Fuzhou, China, **2** Fujian Sanming Forest Ecosystem National Observation and Research Station, Sanming, Fujian, China

* geoyys@fjnu.edu.cn

## Abstract

Subtropical forests are critical to regional carbon cycling, yet the mechanisms by which forest management practices influence soil organic carbon (SOC) stability are not yet fully understood. This study assessed SOC fractions and microbial communities across four forest management types in southeastern China: secondary forest, assisted natural regeneration (ANR), and two monoculture plantations (*Castanopsis carlesii* and *Cunninghamia lanceolata*). We measured soil physicochemical properties, microbial biomass, and community composition, and used structural equation modeling (SEM) to identify the biotic and abiotic pathways regulating SOC dynamics. The results showed that ANR maintained SOC concentrations and microbial traits comparable to those of secondary forests, while significantly reducing labile carbon losses relative to monoculture plantations. At our site, ANR also exhibited higher fine-root input and richer understory cover than plantations, supporting microbially mediated LC retention and, ultimately, greater SOC stabilization. SEM identified two partially independent regulatory pathways: (1) aboveground litter inputs influenced soil pH, carbon:nitrogen ratio, and ammonium nitrogen. These changes promoted the conversion of labile carbon into recalcitrant carbon and increased total SOC. (2) fine root biomass enhanced labile carbon accumulation by increasing the abundance of Gram-positive bacteria. Recalcitrant carbon is conventionally considered a stable SOC component. However, the SEM identified no direct pathway of influence from recalcitrant carbon to total SOC. This finding underscores the central role of labile carbon in SOC stabilization in humid subtropical systems. These findings demonstrate that ANR mitigates carbon losses by enhancing microbially mediated carbon retention and reducing disturbance. This approach supports SOC preservation through improved litter dynamics and root–microbe interactions, providing insights for sustainable forest carbon management.

**Data availability statement:** All relevant data are within the manuscript.

**Funding:** This work was supported by the National Natural Science Foundation of China (grant numbers 32192433, 32101495, and 31930071). Role of the Funder Statement: The funders had no role in study design, data collection and analysis, decision to publish, or preparation of the manuscript.

**Competing interests:** The authors have declared that no competing interests exist.

## Introduction

Soil organic carbon (SOC) represents a major component of terrestrial carbon storage and plays a critical role in global carbon cycling [1]. Forest ecosystems, particularly in subtropical regions, are important carbon reservoirs and have been widely recognized as dynamic systems where SOC turnover is strongly influenced by vegetation type, litter input, and soil microbial processes [2–4]. Forest management can significantly affect soil biogeochemistry and microbially mediated carbon dynamics by altering above- and belowground inputs [5,6]. Traditionally, monoculture plantations have been implemented across subtropical China for timber production. However, such practices have been associated with biodiversity loss, soil degradation, and SOC depletion [7,8]. Assisted natural regeneration (ANR) is a low-cost forest restoration method that minimizes disturbance by retaining logging residues and promoting natural regeneration [5]. It has been widely used in southeastern China, West Africa, and Latin America. ANR enhances multiple ecosystem services, including soil carbon sequestration, biodiversity conservation, hydrological regulation, and improved soil nutrient retention and stability [9–11]. Despite growing interest in ANR, the underlying mechanisms by which it influences SOC accumulation and stabilization, especially through interactions between microbial communities and soil properties, are not well characterized.

Recent studies have highlighted the importance of differentiating SOC into functional fractions, such as labile carbon (LC) and recalcitrant carbon (RC), to better understand the stability and dynamics of soil carbon pools [12,13]. LC, derived from microbial biomass, root exudates, and partially decomposed litter, is sensitive to management-induced disturbance and readily responds to microbial activity [14]. In contrast, RC, which includes chemically complex compounds such as lignin and cellulose, is primarily stabilized through physical protection and mineral association [15]. Therefore, evaluating changes in the relative proportions of LC and RC may better reflect soil carbon pool responses to forest management practices compared to SOC concentration measurements. Such differentiation can facilitate improved soil resource use and ecosystem management [9]. The accumulation, decomposition, and dynamic changes of soil carbon fractions constitute a complex process involving multiple mechanisms [10]. Previous studies have identified influences from various processes, such as litter addition [12], soil and water erosion [13], forest microclimate [16], management disturbances [15], soil physicochemical properties, and biodiversity [17]. Nevertheless, how these drivers vary across management regimes in subtropical forests remains underexplored.

Since SOC fraction dynamics are ultimately governed by microbial-mediated processes, understanding how microbially mediated environmental filters shape microbial functional guilds is critical to deciphering carbon stabilization mechanisms [18,19]. Changes in carbon input and soil nutrient availability under different management measures can affect the composition of microbial communities, especially the balance between bacteria and fungi, and thus affect the decomposition and stability of SOC components [20]. As demonstrated by Lange et al. [21], plant diversity and

plant residue inputs can maintain SOC stocks by stimulating microbial growth. Therefore, enhancing understory vegetation and retaining logging residues during forest management may improve substrate availability for microorganisms, thereby helping preserve microbial community diversity and structural stability [22]. Although previous studies have recognized the role of microbial communities in driving SOC fraction dynamics [23], the mechanisms through which different management practices influence the structural characteristics of soil microbial communities and their relative importance in explaining SOC fraction variations are still unclear.

The subtropical region is characterized by warm and humid conditions with concentrated seasonal rainfall, Its soils are highly weathered, acidic, and nutrient-poor [24]. These edaphic and climatic features not only constrain plant productivity but also render soils more susceptible to erosion and degradation. Together, these factors directly influence SOC dynamics [13]. In addition, frequent transitions between secondary forests and plantations, often accompanied by intensive anthropogenic disturbances such as site preparation and thinning, further impact the stability and accumulation of SOC [25]. Consequently, subtropical forest ecosystems provide an ideal context for exploring how forest management practices regulate soil carbon storage through biotic and abiotic pathways.

To address these questions, we investigated four forest types in a typical subtropical region of southern China: a secondary forest (SF), an ANR forest, a *Castanopsis carlesii* plantation (CCP), and a *Cunninghamia lanceolata* plantation (CLP). Specifically, we aimed to: (1) compare SOC fractions and microbial community composition across different forest management types; and (2) quantify the biotic (microbial traits) and abiotic (soil properties) pathways that drive LC and RC accumulation using structural equation modeling. This study provides new insights into the dual regulatory mechanisms of SOC stability and supports ANR as a viable strategy for carbon conservation and sustainable forest management. We hypothesize that: (1) ANR enhances soil carbon storage by maintaining higher LC and supporting microbial community stability; and (2) monoculture plantations, especially *Cunninghamia lanceolata (CLP)*, reduce SOC concentrations through losses LC and shifts in microbial structure toward oligotrophic groups due to increased disturbance and substrate limitation.

## Materials and methods

### Site description

The study was conducted at a state-owned Chenda Forest Farm, located in Sanming City, Fujian Province, China (117°36′37″E, 26°20′0.8″N; Fig 1). The terrain is predominantly composed of low hills and ridges, with an average elevation of 330 m and a mean slope of 32°. The region experiences a typical subtropical monsoon climate, with a mean annual temperature of 19.1 °C, annual precipitation of 1749 mm (mainly from March to September), annual evaporation of 1585 mm, and relative humidity of 81%. The dominant soil type is Ultisol derived from biotite granite, with a sandy clay texture and bulk density of 1.04 g·cm$^{-3}$. According to the FAO classification system, the soil is classified as Ferric Acrisol, equivalent to Hapludults in the USDA Soil Taxonomy [26].

### Plot design and sampling

In 2011, we established 12 standard observation plots (20 m × 20 m each; Fig 1c) in a SF dominated by *Castanopsis carlesii*, with a shared land-use history and similar slope orientation and gradients to minimize potential impacts of topographic heterogeneity on soil carbon dynamics and microbial community structure. The plots were randomly assigned to four forest management types (each with three replicates): natural secondary forest(SF), ANR dominated by *Castanopsis carlesii*, *Castanopsis carlesii*, CCP and CLP. SF plots (Fig 1c: P10-P12) were control plots with no active management since 1976, allowed to regenerate naturally for over 40 years prior to sampling in 2018. The dominant canopy species were *Castanopsis carlesii*, *Schima superba*, and *Elaeocarpus decipiens*, while the shrub and herb layers primarily comprised *Itea chinensis*, *Symplocos caudata*, *Adinandra millettii*, *Gahnia tristis*, *Dicranopteris dichotoma*, *Ilex pubescens*, and

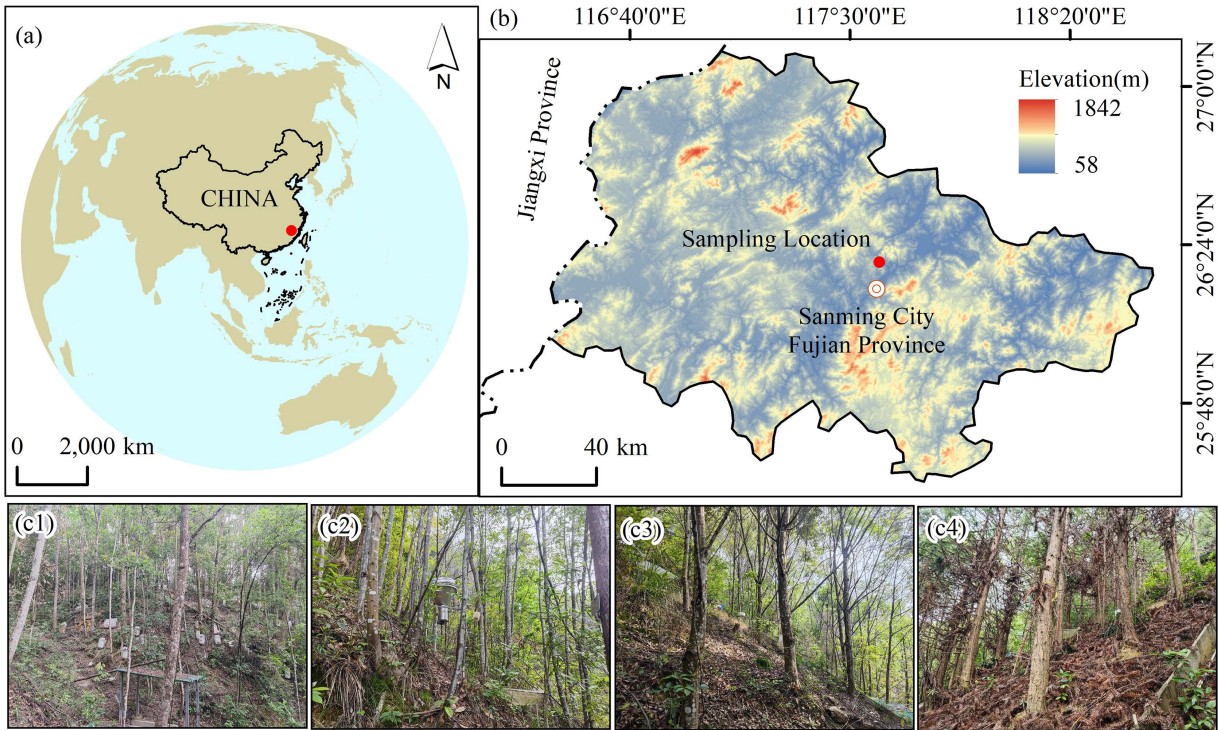

**Fig 1. Location of the study area and representative forest types.** Note: (a) Geographical location of the study site in southeastern China (red dot), generated using QGIS v3.28 with base layers sourced from Natural Earth (https://www.naturalearthdata.com/), a public domain cartographic data source. (b) Topographic features and elevation gradient of the study region in Sanming City, Fujian Province, visualized using SRTM DEM data (NASA Shuttle Radar Topography Mission, https://earthexplorer.usgs.gov/) and administrative boundaries from GADM database (https://gadm.org/). (c) Representative photos of four forest management types. c1: natural SF dominated by *Castanopsis carlesii*; c2: ANR dominated by *Castanopsis carlesii*; c3: *CCP*; c4: CLP.

*Woodwardia japonica* [5]. ANR plots (Fig 1c: P1, P2, P9) followed clear-cutting in December 2011, logging residues were retained in situ to promote natural regeneration of *Castanopsis carlesii* seedlings, with minimal subsequent disturbances. CCP plots (Fig 1c: P3, P4, P8) and CLP plots (Fig 1c: P5, P6, P7) underwent controlled burning in March 2012 prior to *Castanopsis carlesii* and *Cunninghamia lanceolata* seedlings being planted. Regular shrub removal was performed in both plantation types to maintain consistent management practices.

Because surface soils represent hotspots of microbial activity and store the majority of SOC pool [27,28]. In June 2018, we selected 15 random points in each plot using an S-shaped sampling pattern to collect soil cores from the 0–10 cm layer. The homogenized soil cores were stored in a portable 12 V refrigerator maintained at 4°C immediately after field collection, and transported to the laboratory within 24h to minimize microbial activity changes prior to analysis. After removing roots and gravel, one portion of each sample was passed through a 2 mm sieve for phospholipid fatty acids (PLFA) analysis, while the remainder was air-dried and sieved to 0.149 mm for measuring SOC fractions, and carbon (C), nitrogen (N), and phosphorus(P) contents.

## Separation of labile and recalcitrant organic carbon fractions

Although the acid hydrolysis method [29] may result in overlaps between labile and recalcitrant carbon pools, it remains widely used to evaluate relative changes in SOC fractions under different management practices due to its efficiency, simplicity, and reproducibility. We implemented the acid hydrolysis procedure as follows: (1) We added 500 mg of air-dried

soil to 20 mL of 2.5 mol $L^{-1}$ $H_2SO_4$ and boiled at 105 °C for 30 min. (2) The slurry was transferred to a 50 mL centrifuge tube and centrifuged at 4500 rpm for 30 min; the supernatant was decanted. (3) We added 30 mL ultrapure water, centrifuged again, combined the two supernatants, and filtered through a 0.45 μm membrane. This filtrate (starches and soluble sugars) was defined as labile carbon fraction 1 (LC1). (4) The residue was rinsed repeatedly with deionized water and dried at 60 °C to constant mass. We then added 2 mL of 13 mol $L^{-1}$ $H_2SO_4$, shook for 10 h, diluted to 1 mol $L^{-1}$ $H_2SO_4$, and boiled at 105 °C for 3 h. (5) After boiling, we centrifuged at 4000 rpm for 10 min, decanted the supernatant, rinsed the residue with 20 mL distilled water, combined supernatants, and filtered through a 0.45 μm membrane. This filtrate (cellulose and other water-soluble compounds) was defined as LC2. (6) Total LC was the sum of LC1 and LC2. The final residue was rinsed to neutrality and dried to constant mass at 60 °C; this residue was considered recalcitrant carbon (RC) (mainly lignin, humic and fulvic substances). (7) LC was quantified using a total organic carbon analyzer, and total soil C and RC were determined using a C–N analyzer (Elementar Vario MAX, Hanau, Germany).

## Determination of soil physicochemical properties and microbial community structure

Soil moisture (SM) was determined by the drying method. SOC and total nitrogen (TN) were measured using a C–N analyzer. Soil pH was determined by the glass electrode method (soil:water = 1:2.5) after 30 min of shaking and 1h of standing, using a pH meter (Leici PHSJ-4F, China)). Mineral nitrogen ($NH_4^+$-N and $NO_3^-$-N) was extracted with 2 mol $L^{-1}$ KCl and determined using a continuous flow analyzer (Skalar San++, Holland). Available phosphorus (AP) was extracted using the Mehlich 3 method, filtered with a phosphorus-free filter paper, and determined using a continuous flow analyzer (Skalar San++, Holland). Total phosphorus (TP) was digested with $H_2SO_4$–$HClO_4$ and determined using an inductively coupled plasma emission spectrometer (PerkinElmer 7000DV, USA). The hydrometer method was used to determine the clay, silt, and sand content of the soil samples. Litter stock was quantified within a 50 × 50 cm quadrat, oven-dried at 65°C to constant mass, and expressed as Mg·ha⁻¹. Fine root biomass (diameter ≤ 2 mm) was determined by soil coring (0–10 cm depth), followed by washing through a 0.15 mm sieve, drying at 65°C for 48 h, and reported per unit are(Mg·ha⁻¹).

The microbial community structure of the soil was determined using PLFAs. The concentrated sample was transferred to a silinized tube, and the molecular structure of fatty acids was obtained using gas chromatography (Agilent 6890 N, USA) combined with the MIDI microbial identification system (MIDI Inc., Newark, DE). Standard nomenclature was used to further differentiate microbial groups. PLFAs i-14:0, i-15:0, a-15:0, i-16:0, i-17:0 and a-17:0 were considered representative of Gram-positive bacteria (GP), while 16:1 ω9c, 16:1 ω7c, cy-17:0 ω7c, 18:1 ω7c and cy-19:0 ω7c were representative of Gram-negative bacteria (GN). The PLFAs 18:2 ω6c and 18:1 ω9c were representative of fungal species (FUNGI), and the PLFAs 16:0 10-methyl, 17:0 10-methyl, 18:0 10-methyl, 17:1 ω7c 10-methyl and 18:1 ω7c 10-methyl were representative of actinomycetes (ACT). The PLFAs 16:1 ω5c was representative of arbuscular mycorrhizal fungi (AMF). The ratio of fungal to bacterial PLFAs(F:B) was used to estimate the relative importance of bacterial and fungal metabolism in the community [30].

## Statistical analysis

We used Kolmogorov-Smirnov tests and Levene's tests to assess data normality and homogeneity of variances, respectively. Group differences in SOC fractions, soil physicochemical properties, and microbial community variables among forest management types were determined using one-way ANOVA with Tukey's post-hoc comparisons (α = 0.05). To elucidate how forest management affects SOC dynamics, we first conducted Mantel tests using the linkET package (v0.0.8) in R 4.3.1 (R Core Team, 2023), evaluating multivariate correlations between management-induced environmental gradients and carbon component matrices. Subsequently, a structural equation model (SEM) was constructed based on a priori ecological hypotheses. Model fitting was performed using the lavaan package (v0.6-16) with maximum likelihood estimation. Model fit was assessed through three criteria: $\chi^2$/df ratio, root mean square error of approximation (RMSEA), and

comparative fit index (CFI). Nonsignificant paths ($P \geq 0.05$) were iteratively removed, while biologically plausible pathways with modification indices (MI) >10 were incorporated to optimize model structure. Standardized path coefficients quantified directional effect magnitudes between variables, with $R^2$ values indicating the proportion of variance explained for each endogenous variable.

## Results

### Differences in soil physicochemical properties

As shown in Table 1, TP, pH, SM, and clay content did not differ significantly among forest types ($P > 0.05$). By contrast, nitrogen dynamics and carbon-allocation metrics diverged across forests, indicating management effects on active C and N pools. Specifically, the highest concentrations of TN and $NH_4^+$-N were recorded in SF ($P < 0.05$), while $NO_3^-$-N in SF was 12% of that in ANR ($P < 0.05$). In contrast, both CCP and CLP exhibited significantly reduced $NH_4^+$-N compared to SF and ANR ($P < 0.05$).

ANR displayed the highest C:N ratio ($P < 0.05$), which sharply contrasted with plantations (18.29–19.54; $P < 0.05$). Plant-derived inputs also diverged. ANR accumulated 73% more litter stock(LS) than SF ($P < 0.05$), exceeding plantations by 2.7–3.7 ($P < 0.05$). SF and ANR retained comparable fine root biomass (2.14–2.28 Mg ha$^{-1}$; $P > 0.05$), both significantly surpassing plantations (1.27–1.79 Mg ha$^{-1}$; $P < 0.05$).

### Differences in SOC and its fractions

Significant alterations in SOC characteristics and its fractions were observed in the 0–10 cm soil layer under different forest management regimes (Fig 2). The highest SOC concentration was recorded in SF (26.31 ± 1.03 g·kg$^{-1}$), with no significant difference detected between ANR (23.89 ± 0.76 g·kg$^{-1}$) and SF ($P > 0.05$). In contrast, SOC was lower in CCP and CLP by 21.04% and 30.44%, respectively (both P < 0.05). Total LC declined progressively across management types relative to SF (−14.50% to −46.10%; P < 0.05). The rapidly decomposable fraction (LC1) decreases of 22.62% (ANR), 30.19%(CCP), and 44.83%(CLP) ($P < 0.05$). RC in CLP (9.18 ± 0.56 g·kg$^{-1}$) was significantly lower than in of SF (14.29 ± 1.26 g·kg$^{-1}$), ANR (12.72 ± 0.43 g·kg$^{-1}$), and CCP (13.20 ± 0.56 g·kg$^{-1}$) ($P < 0.05$).

**Table 1.  Variations in basic soil physicochemical properties among forest types (n = 3).**

| Forest types | TP (mg·kg$^{-1}$) | TN (g·kg$^{-1}$) | $NH_4^+$-N (mg·kg$^{-1}$) | $NO_3^-$-N (mg·kg$^{-1}$) | C:N |
|---|---|---|---|---|---|
| SF | 162.5 ± 9.3a | 1.32 ± 0.01a | 17.61 ± 0.12a | 0.32 ± 0.04c | 19.88 ± 0.78ab |
| ANR | 167.6 ± 12.6a | 1.08 ± 0.04b | 17.09 ± 1.17a | 2.7 ± 0.41a | 22.29 ± 1.31a |
| CCP | 148.1 ± 14a | 1.06 ± 0.03b | 10.26 ± 1.91b | 1.06 ± 0.09b | 19.54 ± 0.18b |
| CLP | 150.2 ± 2.5a | 1 ± 0.08b | 7.04 ± 1.63b | 0.38 ± 0.14bc | 18.29 ± 0.62b |
| Forest types | pH | SM (%) | Clay (%) | LS (Mg ha$^{-1}$) | FRB (Mg ha$^{-1}$) |
| SF | 4.87 ± 0.11a | 10.19 ± 1.74a | 10.61 ± 1.26a | 3.62 ± 0.24b | 2.14 ± 0.03a |
| ANR | 4.68 ± 0.09a | 9.99 ± 2.91a | 10.5 ± 0.91a | 6.26 ± 0.51a | 2.28 ± 0.06a |
| CCP | 4.78 ± 0.03a | 7.69 ± 1.73a | 12.4 ± 0.53a | 4.24 ± 0.22b | 1.79 ± 0.05b |
| CLP | 4.74 ± 0.16a | 12.29 ± 1.66a | 11.84 ± 1.17a | 1.7 ± 0.56c | 1.27 ± 0.05c |

Note: Values are expressed as mean ± standard error (SE). Different lowercase letters indicate significant differences at α = 0.05 level. SF: natural secondary forest; ANR: assisted natural regeneration forest of *Castanopsis carlesii*; CCP: *Castanopsis carlesii* plantation; CLP: *Cunninghamia lanceolata* plantation; TP: total phosphorous; TN: total nitrogen; C:N: carbon nitrogen ratio; SM: soil moisture; LS: litter storage; FRB: fine root biomass; $NH_4^+$-N: ammonium nitrogen; $NO_3^-$-N: nitrate nitrogen; pH: soil pH. The same below.

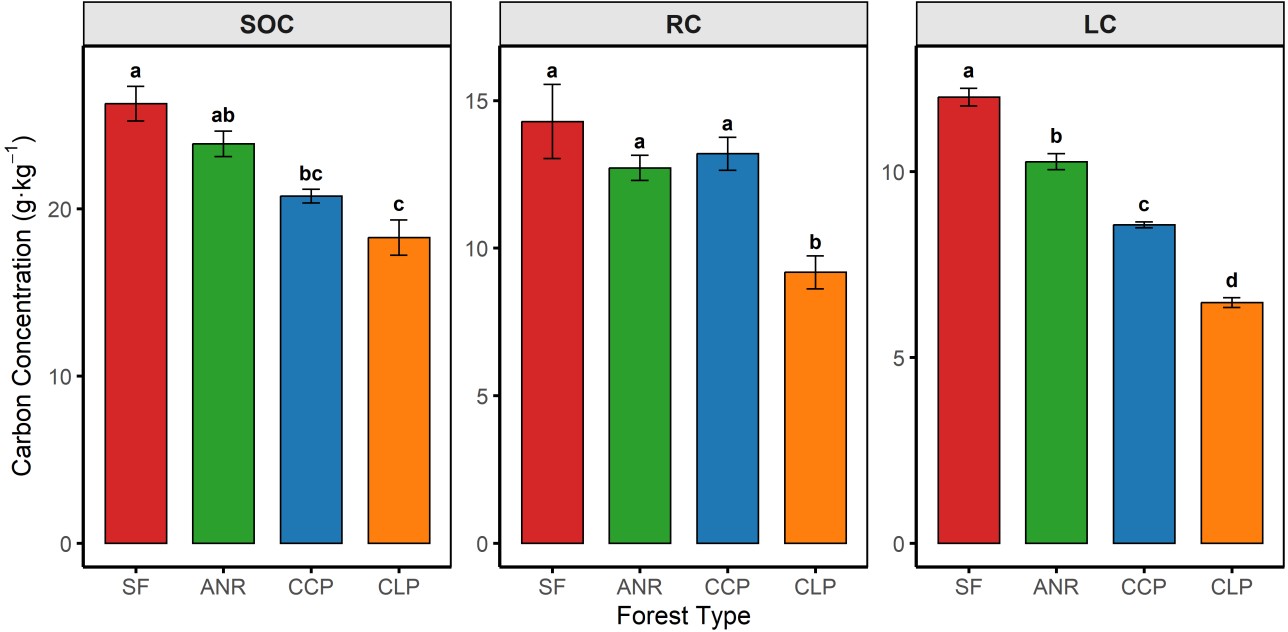

**Fig 2. Effects of different management practices on soil organic carbon (SOC) and its fractions.** Note: LC: labile carbon fractions; RC: recalcitrant carbon. SF: natural secondary forest; ANR: assisted natural regeneration forest of *Castanopsis carlesii*; CCP: *Castanopsis carlesii* plantation; CLP: *Cunninghamia lanceolata* plantation. Different lowercase letters indicate significant differences between different forest management practices (n = 3). The same below.

## Soil microbial community composition

Significant alterations in soil microbial community structure were observed under different forest management practices (Table 2). Total PLFA concentrations showed no significant difference between SF and ANR ($P > 0.05$), but were 42.4%–45.0% lower in plantations (CCP, CLP) than in SF ($P < 0.05$).

Among functional guilds, GN and GP bacteria showed parallel declines in plantations(CCP and CLP: −52.9% and −39.5%, respectively; $P < 0.05$). Fungal abundance in ANR was 6.6% lower than SF (ns), but declined by 25.9–36.3% in plantations, with a significant decrease in CCP ($P < 0.05$). ACT and AMF PLFAs were also reduced in plantations (ACT

**Table 2. PLFAs characteristics of soil microorganisms under different forest management practices.**

| Microorganism PLFAs (mol·g⁻¹) | SF | ANR | CCP | CLP |
|---|---|---|---|---|
| Total | 42.43 ± 3.05a | 36.89 ± 2.92a | 23.27 ± 3.88b | 23.34 ± 1.78b |
| GN | 14.34 ± 1.01a | 11.94 ± 0.66a | 6.76 ± 1.53b | 6.83 ± 0.91b |
| GP | 12.85 ± 0.87a | 10.54 ± 0.77a | 7.77 ± 1.15b | 7.17 ± 0.21b |
| FUNGI | 7.19 ± 0.54ab | 7.67 ± 1.19a | 4.59 ± 0.83b | 5.33 ± 0.73ab |
| ACT | 6.04 ± 0.52a | 4.93 ± 0.39a | 3.28 ± 0.49b | 2.98 ± 0.22b |
| AMF | 2.01 ± 0.12a | 1.82 ± 0.153a | 0.87 ± 0.28b | 1.03 ± 0.15b |
| GP:GN | 0.89 ± 0.01b | 0.88 ± 0.02b | 1.21 ± 0.18a | 1.09 ± 0.14ab |
| F:B | 0.28 ± 0.01b | 0.34 ± 0.04ab | 0.31 ± 0.02ab | 0.38 ± 0.04a |

Note: Different lowercase letters indicate statistical significance among different forest management practices at α = 0.05. The data in the table are presented as mean ± standard error with n = 3. PLFAs: phospholipid fatty acids; GN: Gram-negative bacteria; GP: Gram-positive bacteria; FUNGI: fungal species; ACT: actinomycetes; AMF: arbuscular mycorrhizal fungi; F:B: the ratio of fungal to bacterial PLFAs.

−45.7–50.7%; AMF −48.6–56.7%; P<0.05). Community indices increased from GP:GN=0.88–0.89 in SF/ANR to 1.09–1.21 in plantations (P<0.05), indicating a relative dominance of GP under managed regimes. The F:B was significantly higher in CLP than in SF (P<0.05). A plausible ecological context is that litter quality and rooting traits differ between broad-leaved *Castanopsis* (CCP) and coniferous *Cunninghamia* (CLP); slower-decomposing, nutrient-poorer litter and reduced belowground inputs in CLP likely constrained microbial biomass and shifted community structure.

## Relationships between SOC, microbial characteristics, and soil properties

As shown in Fig 3, SOC exhibited the strongest associations with belowground biomass and N-availability: fine root biomass (FRB), TN, and $NH_4^+$-N (all P<0.001). SOC was also correlated with GP, GN, AMF, ACT, and total PLFAs(P<0.05). In contrast, RC related to fewer factors—most strongly to FRB (P<0.001)—and additionally to LS and $NH_4^+$-N (P<0.05), with weaker links to other soil properties. LC was regulated by multiple variables, showing strong correlations (P<0.001) with total PLFAs, ACT, AMF, GP, GN, $NH_4^+$-N, TN, and FRB, and a significant correlation with LS (P<0.005). Notably, pH, clay, SM, F:B, and $NO_3^-$-N showed weak associations, suggesting limited direct contributions or mediation via other drivers.

As illustrated in Fig 4, the SEM elucidates two ecologically meaningful pathways. First, greater litter inputs increased the C:N ratio and lowered pH, which together promoted higher $NH_4^+$-N and, in turn, the conversion of LC to RC and

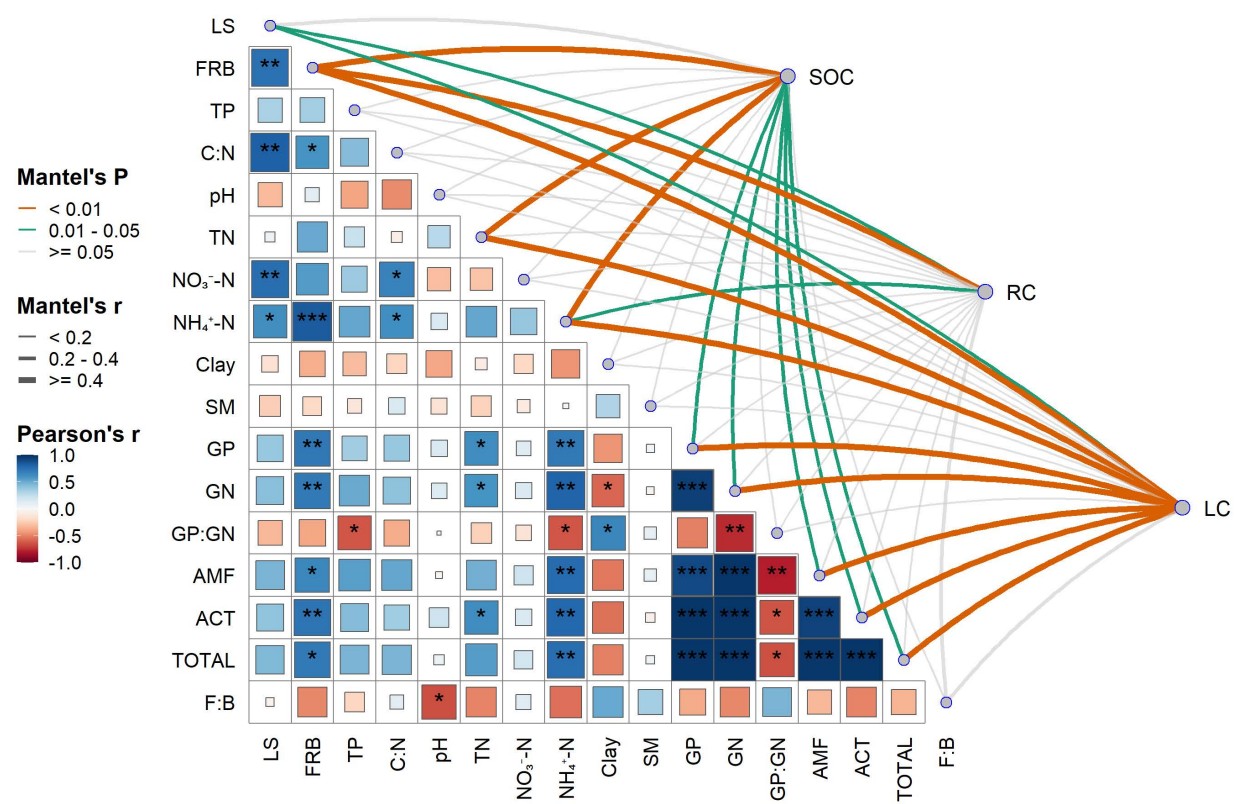

**Fig 3. Environmental drivers of SOC and its fractions.** Note: The figure shows pairwise comparisons between microbial and soil property indicators, with color gradient indicating the Pearson correlation coefficient. SOC: soil organic carbon; RC: recalcitrant carbon; LC: labile carbon; TN: total nitrogen; Ammonium. N and Nitrate. N: mineral nitrogen; TP: total phosphorus; Clay represents the clay percentage; SM: soil moisture. Using Mantel tests, the correlations between SOC and its fractions (LC, RC) with each environmental factor were determined. The line color represents the significance level of Mantel's P value, and the line width corresponds to the Mantel's r value.

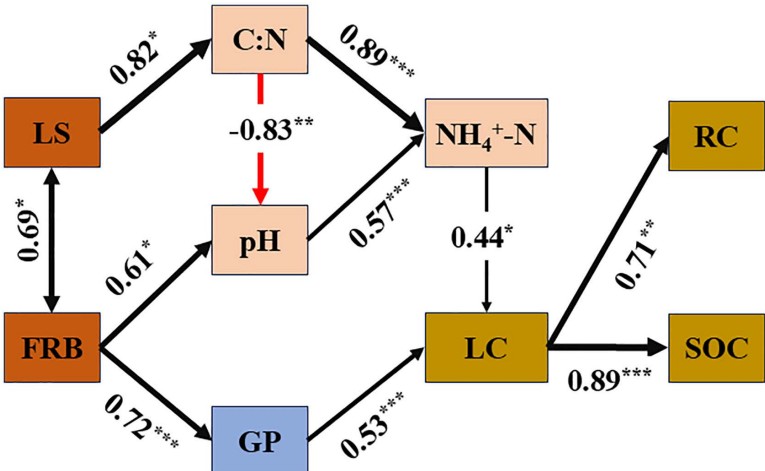

**Fig 4. Scores and interrelationships of the best predictors for predicting SOC.** Note: Black arrow shows positive influence, red arrow shows negative influence, the value on each arrow lines are the path coefficient. * presents significant at $P < 0.05$, ** presents significant at $P < 0.01$, *** presents significant at $P < 0.001$. SOC: soil organic carbon; RC: recalcitrant carbon; LC: labile carbon; GP: Gram-positive bacteria; C:N: carbon nitrogen ratio; LS: litter storage; FRB: fine root biomass.

higher total SOC. Second, larger FRB increased the abundance of GP, enhancing LC accumulation. Consistent with these pathways, LC strongly predicted SOC, whereas no direct path from RC to SOC was detected. Model statistics ($\chi^2$/df, RMSEA, CFI; see caption) indicate acceptable fit given sample size (n = 12).

## Discussion

### Impacts of Forest management on soil physicochemical properties and microbial characteristics

In this study, different forest management practices significantly influenced TN, mineral nitrogen, and C:N, whereas their effects on TP, SM, pH, and clay content were relatively minor (Table 1). LS was found to be positively correlated with soil C:N and ammonium nitrogen, but negatively correlated with pH (Fig 3). Previous studies have indicated that forest management can directly improve soil nutrient status by altering understory vegetation composition [31]. Consistent with this view, SEM results (Fig 4) further revealed that LS was a key driver of variations in soil C:N, ammonium nitrogen, and pH. Although broadleaf species were abundant in both SF and ANR forests, the soil C:N in these forests was higher than that in the CCP and CLP monoculture plantations. While such a pattern is uncommon, it has been reported elsewhere. For instance, Lin et al. [32]observed that birch forests exhibited higher soil C/N compared to coniferous forests, and similar trends have been documented in Iceland and Finland [33], where broadleaf forests showed higher C:N ratios than coniferous stands. These differences may be attributed to variations in litter decomposition rates and seasonal sampling. Broadleaf litter tends to have higher nitrogen content and decomposes rapidly during early stages. However, as recalcitrant components accumulate over time, the decomposition rate gradually slows and eventually falls below that of coniferous litter, leading to an increase in soil C:N. Additionally, during the growing season, SF and ANR forests typically exhibit higher nitrogen uptake to support leaf development, as observed by Bradley [34], who reported increased nitrogen absorption in soils supporting actively growing seedlings. Seasonal sampling and differences in decomposition stages further explain the elevated levels of TN and ammonium nitrogen observed in the secondary and ANR forests. Moreover, nitrogen availability is a critical factor influencing the distribution of soil carbon fractions. Studies by Wang [31] and Wu et al. [35] have shown that higher C:N ratios promote the accumulation of LC fractions, supporting the SEM-derived pathway in this study, whereby LS affects labile carbon accumulation via modulation of soil C:N. Direct plot measurements (Table 3) support

**Table 3. Characteristics of sample plots.**

| Forest management types | Mean slope (°) | Mean height (m) | Mean DBH (cm) | Mean coverage | Fine root yield (g·m⁻²·yr⁻¹) | Average annual decline of biomass (t·ha⁻¹) |
|---|---|---|---|---|---|---|
| SF | 32.7 | 19.7 | 13.5 | 93.3% | 75.2 | 7.0 |
| ANR | 31.7 | 7.2 | 5.8 | 92.7% | 284.0 | 4.6 |
| CCP | 32.2 | 6.4 | 8.5 | 84.5% | 163.0 | 2.9 |
| CLP | 32.4 | 9.2 | 12.6 | 82.1% | 144.6 | 2.4 |

Note: SF: natural secondary forest; ANR: assisted natural regeneration forest of *Castanopsis carlesii*; CCP: *Castanopsis carlesii* plantation; CLP: *Cunninghamia lanceolata* plantation; DBH: Diameter at Breast Height.

stronger belowground inputs and a richer understory under ANR: fine-root yield is 284 g m⁻² yr⁻¹ in ANR vs 163 in CCP and 145 in CLP, and understory cover is 92.7% in ANR (93.3% in SF) vs 84.5%(CCP) and 82.1%(CLP). These features likely increase substrate supply to microbes and help sustain LC pools and microbial biomass relative to plantations (Table 3).

Due to intrinsic differences between fungi and bacteria in enzyme activity [36], biomass turnover rate [37], and carbon use efficiency [38], microbial community composition has been shown to respond differently to forest management regimes [39]. In this study, microbial biomass in the ANR and secondary forests was comparable and significantly higher than that in CCP and CLP. In contrast, both plantations exhibited elevated fungal-to-bacterial and GP:GN ratios (Table 2), consistent with the findings of Zhao et al. [27] and Wan et al.[40]. These results suggest that ANR management enhanced microbial biomass by increasing substrate availability, likely through fineroot biomass accumulation [41,42]. In contrast, common plantation practices such as litter burning and understory removal reduced both above- and belowground carbon inputs [43], thereby intensifying microbial resource limitations [44]. In plantations, higher F:B and GP:GN likely reflect adaptation to high C:N substrates and nutrient limitation [24]. The muted contrasts between CCP and CLP may relate to stand age, with communities not yet at a new equilibrium [45].

## Shifts in microbial functional composition

Subtropical forests in China play a vital role in regional carbon cycling. Previous studies have demonstrated that, compared to monoculture plantations, both secondary forests and ANR practices contribute to the accumulation of SOC [25,28,46], a pattern that was further confirmed in this study (Fig 2). Beyond confirming this pattern, we clarify how management regulates SOC via coordinated responses of LC and RC to above-/belowground inputs. These results complement earlier findings on ANR`s role in SOC retention [25,46,47]. The results showed that, compared to the secondary forest, the LC fractions in ANR, CCP, and CLP forests decreased by 14.49%, 28.64%, and 46.13%, respectively. However, the LC loss in the ANR forest was substantially lower than that observed in the two monoculture plantations (Fig 2), consistent with previous findings [47]. Because LC derives from easily decomposable substrates and is disturbance-sensitive [48], continuous litterfall and root exudates in SF/ANR likely sustain LC pools [5], whereas early site preparation and understory removal in plantations reduce inputs and deplete LC [48–50].

Regarding the RC fraction, higher contents were observed in the SF and ANR forests (Fig 2), which may be attributed to enhanced environmental stability associated with reduced disturbance and long-term litter accumulation. In contrast, the RC content in the CLP was the lowest, with a 35.76% reduction compared to SF. No significant difference in RC was detected between the CCP and SF or ANR forests, which is consistent with previous studies [48–50]. RC, RC, composed of chemically complex residues (cellulose, hemicellulose, lignin), underpins long-term stabilization of SOC. Although both plantation types were subjected to early site disturbances that reduced litter input and increased aggregate oxidation [28], RC loss in the CCP was not significant, Th possibly because higher-quality Castanopsis litter more readily forms stable mineral associations [51]. In contrast, slower litter decomposition and nutrient return in the *Cunninghamia lanceolata*(CLP)

plantation may have limited RC formation and stabilization, yielding the lowest RC. The overall fit statistics ($\chi^2 = 35.91$, df = 24, P = 0.056; RMSEA = 0.203) indicate marginal global misfit but an informative structure. Ecologically, this means the hypothesized pathways capture the dominant processes—(i) greater litter inputs elevating C:N and lowering pH, which increase $NH_4^+$-N and promote LC→RC conversion and higher SOC; and (ii) larger FRB increasing GP abundance and boosting LC—while the elevated RMSEA cautions that additional, unmeasured processes (e.g., microclimate or substrate chemistry) may also contribute given n = 12.

## Influence Pathways of Forest Management on SOC Pool

Studies have shown that different forest management practices can regulate both above- and belowground carbon inputs, thereby triggering coupled changes in soil environmental conditions and microbial community structure, ultimately affecting the accumulation and distribution of SOC [52]. Previous research has generally suggested that SOC dynamics are primarily driven by microbial community composition, with soil environmental factors playing only an indirect role. However, SEM constructed in this study (Fig 4) demonstrated that soil physicochemical properties and microbial biomass contributed equally to the regulation of the LC fraction and, in turn, to SOC and RC accumulation.

Correlation analysis revealed that microbial community structure and abundance were primarily influenced by ammonium nitrogen concentrations and fine root biomass, while TP and pH also exhibited significant effects on microbial indices such as the GP:GN ratio and F:B (Fig 3). However, no significant path relationships were observed between microbial indicators and basic soil physicochemical properties (Fig 4). Among these, the abundance of GP bacteria was positively affected only by root biomass. These findings suggest that the differences in SOC among the four forest management regimes were regulated by two relatively independent pathways. On one hand, variations in aboveground litter input induced by management practices primarily influenced soil pH, C:N ratio, and ammonium nitrogen content, thereby regulating the transformation from LC to RC and overall SOC accumulation. On the other hand, changes in belowground FRB affected microbial community structure, particularly by increasing the proportion of Gram-positive bacteria, which in turn promoted LC accumulation. This mechanism is consistent with the findings of Drum et al. [53]. Similarly, studies by Song [54] and DuPont [55] reported that root exudates, as sources of substrate input, can significantly enhance microbial activity and biomass, thereby expanding the LC pool in surface soils and strengthening soil carbon sequestration capacity. Although an increase in LC is often considered a potential risk to SOC stability due to its association with enhanced decomposition and transformation rates [50], our SEM indicates that LC mediates the buildup of both RC and total SOC, and favorable conditions in SF/ANR (greater diversity, higher litter quality, lower disturbance) likely facilitate LC→RC conversion and long-term carbon retention. It is noteworthy that no significant path was detected between RC and SOC in the model, which contrasts with the findings of Liu et al. [56] in a semi-arid agro-pastoral ecotone. This discrepancy suggests that in humid subtropical ecosystems, characterized by active biogeochemical cycling, LC plays a more critical role in maintaining SOC stability [57]. Therefore, sustaining LC levels may be essential for mitigating SOC loss under such conditions. This mechanism corroborates our first hypothesis by demonstrating that LC accumulation and its conversion to RC under ANR management are key processes supporting SOC stability.

## Ecosystem implications and management recommendations

Our findings support the hypothesis that ANR offers a sustainable alternative to conventional plantations by maintaining microbial structure and enhancing LC retention. The ability of ANR to promote LC→RC transformation, likely via improved litter quality and reduced disturbance, underscores its role in long-term SOC stabilization. Given that microbially mediated LC transformation is a dominant driver of SOC accumulation in these systems, forest policies aiming at carbon sequestration should prioritize strategies that preserve microbial diversity and substrate availability. Furthermore, the observed decoupling of RC and SOC underlines the importance of monitoring LC dynamics in subtropical soils. While our study

revealed key patterns in SOC fraction regulation, long-term monitoring and broader site comparisons are needed to fully understand temporal dynamics and broader applicability of ANR practices across varying subtropical regions.

## Conclusions

This study provides empirical evidence on the divergent impacts of subtropical forest management practices on soil organic carbon (SOC) fractions and their microbially mediated regulatory mechanisms. Assisted natural regeneration (ANR), by minimizing anthropogenic disturbances and preserving natural litter input, was shown to sustain SOC concentrations and microbial community structure comparable to those found in secondary forests(SF). Importantly, ANR enhanced the conversion efficiency of labile carbon (LC), supporting carbon retention through a dual regulatory mechanism: (1) aboveground litter influencing soil C:N and $NH_4^+$–N concentrations to promote LC transformation into RC and SOC, and (2) belowground fine root biomass modulating microbial composition to foster LC accumulation. In contrast, monoculture plantations, especially *Cunninghamia lanceolata*, experienced concurrent losses of LC and recalcitrant carbon (RC) and SOC, and (2) belowground fine root biomass modulating microbial composition to foster LC accumulation.

In contrast, monoculture plantations, especially Cunninghamia lanceolata plantations (CLP), experienced concurrent losses of both LC and RC fractions (with SOC reduced by up to 30.4% relative to SF), and exhibited microbial shifts toward oligotrophic dominance—evidenced by increased fungal-to-bacterial and Gram-positive:Gram-negative (GP:GN) ratios—ultimately accelerating SOC decomposition. Structural equation modeling (SEM) indicated that LC serves as a critical mediator for SOC stabilization in humid subtropical systems, while RC alone showed no direct effect on SOC accumulation. This finding highlights the key role of maintaining active LC turnover through biological processes for long-term carbon retention.

## Author contributions

**Conceptualization:** Zhiqiang Ding, Zhijie Yang, Yusheng Yang.

**Data curation:** Zhiqiang Ding, Zhijie Yang.

**Formal analysis:** Zhiqiang Ding, Zhijie Yang, Yusheng Yang.

**Funding acquisition:** Yusheng Yang.

**Investigation:** Zhiqiang Ding.

**Methodology:** Zhiqiang Ding, Zhijie Yang.

**Project administration:** Yusheng Yang.

**Software:** Zhiqiang Ding.

**Supervision:** Zhijie Yang.

**Visualization:** Zhiqiang Ding.

**Writing – original draft:** Zhiqiang Ding, Yusheng Yang.

**Writing – review & editing:** Zhiqiang Ding, Zhijie Yang, Yusheng Yang.

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
