## [Decision Letter · Decision Letter 0]

9 Feb 2025

Dear Dr. Ding,

Thank you for submitting your manuscript to PLOS ONE. After careful consideration, we feel that it has merit but does not fully meet PLOS ONE’s publication criteria as it currently stands. Therefore, we invite you to submit a revised version of the manuscript that addresses the points raised during the review process.

We look forward to receiving your revised manuscript.

Kind regards,

Tunira Bhadauria, Ph.D.

Academic Editor

PLOS ONE

“This work was supported by the National Nature Science Foundation of China (NO. 32192433, 32101495 and 31930071).”

“This work was supported by the National Nature Science Foundation of China (NO. 32192433, 32101495 and 31930071).”

“This work was supported by the National Nature Science Foundation of China (NO. 32192433, 32101495 and 31930071).”

5. We note that your Data Availability Statement is currently as follows: [All relevant data are within the manuscript and its Supporting Information files.]

6. We note that Figure 1 in your submission contain [map/satellite] images which may be copyrighted. All PLOS content is published under the Creative Commons Attribution License (CC BY 4.0), which means that the manuscript, images, and Supporting Information files will be freely available online, and any third party is permitted to access, download, copy, distribute, and use these materials in any way, even commercially, with proper attribution. For these reasons, we cannot publish previously copyrighted maps or satellite images created using proprietary data, such as Google software (Google Maps, Street View, and Earth). For more information, see our copyright guidelines: http://journals.plos.org/plosone/s/licenses-and-copyright.

Reviewers' comments:

Reviewer's Responses to Questions

**Comments to the Author**

1. Is the manuscript technically sound, and do the data support the conclusions?

Reviewer #1: Yes

Reviewer #2: Partly

2. Has the statistical analysis been performed appropriately and rigorously?

Reviewer #1: No

Reviewer #2: Yes

3. Have the authors made all data underlying the findings in their manuscript fully available?

Reviewer #1: Yes

Reviewer #2: Yes

4. Is the manuscript presented in an intelligible fashion and written in standard English?

Reviewer #1: Yes

Reviewer #2: Yes

Reviewer #1: This MS titled “Assisted natural regeneration Forests demonstrate greater carbon storage potential than plantation in subtropical China” aims to explore effects of forest management strategies (i.e., ANR and man-made plantation) on SOC fractions and SMC structures over 6 years. While the topic is relevant and widely studied, the methods and findings presented in this MS appear somewhat ordinary. Additionally, the structure and content require further depth and refinement. Below are detailed suggestions for improvement.

1) this work did not directly address carbon storage, and the term "C storage" in the title should be replaced with "C fractions" to more accurately reflect the study's focus.

2) INTRODUCTION. The introduction lacks sufficient focus on the major topic. More attention should be given to building a strong theoretical basis for the hypotheses, which are currently missing. Clear articulation of hypotheses is essential. Some extraneous information should be removed, such as L80–85 (discussion on rhizosphere and plant and microbial diversity), as they are not directly relevant to your study. The background on ANR (L41–49) is overly detailed and should be condensed to improve clarity and flow.

3) M&M

L120-128, were the treatments randomly assigned?

L190-191, one-way ANOVA is typically used to test the significance of treatment effects on response variables or their difference, not their relationships. please rephrase or clarify the intended purpose of this analysis.

4) the REUSLTS section needs reorganization with more concise and accurate statements focusing on the key findings.

L245-261, the influencing factors (soil physicochemical properties and SMC groups) are insufficient. for example, PH should be included as it plays a critical role in microbial processes and SOC fraction dynamics. Litter and root characteristics, as major sources of SOC and microbes (particularly LC), should be provide and analyze. These are often considered essential when analyzing SOC fractions in forests. In my opinion, physicochemical properties should also be tested across treatments (as with SMC groups), and the significant variables would be considered for further analysis.

5) there are so many spelling, grammar and writing errors that need significant improvement. please ensure a thorough review before submission. For example, L13-14, 18-19, 25-26,,,

L49, the abbreviation of SOC should be labeled when it first mentioned. And also, L57 (CNP), L139 (carbon nitrogen phosphorus) ,,,

Reviewer #2: This manuscript investigates the comparative potential of Assisted Natural Regeneration (ANR) and traditional plantation forests for carbon storage and microbial community modulation in subtropical China. This research topic is interesting and has practical significance. The study employs a robust experimental design with well-replicated field trials, providing valuable insights into the complex interplay between soil carbon dynamics and microbial ecology. However, while the findings are promising, certain methodological and interpretational aspects require further refinement. Specifically, the connection between the results and broader ecological implications could be more clearly articulated, and addressing some inconsistencies in methodology and data presentation would significantly enhance the manuscript's scientific rigor and impact. Furthermore, the discussion sections of the manuscript are the weakest and require further improvement. I have provided specific comments below.

Specific Comments:

Title: The title appears overly broad given the scope of the study. A single study may not sufficiently support the generalization implied in the title. Consider narrowing the title to reflect the specific findings, such as emphasizing the study’s focus on carbon storage and microbial community dynamics in a particular region and forest type.

Abstract: The abstract seems to be a repetition of the results and lacks key supporting data.

Line 12: Rephrase "how it performs than" to "how it performs relative to" for improved fluency and precision.

Line 17: Clearly define "LC1" and "LC2" as "labile carbon fractions with varying turnover times" to enhance precision and accessibility for a multidisciplinary audience.

Introduction: Provide a more nuanced discussion of the subtropical region’s unique climatic and edaphic conditions, emphasizing their relevance to soil carbon storage and forest restoration dynamics.

Line 48: Elaborate on the phrase "extensive ecosystem service functions" by detailing specific services linked to ANR, such as carbon sequestration, hydrological regulation, and biodiversity enhancement.

Lines 118-125: This sentence is too complicated, please simplify it. In addition, please specify whether the plots were randomized or stratified based on biophysical variables like slope, aspect, and vegetation type, as these can significantly influence soil and microbial metrics.

Line 135: Justify the selection of a 0-10 cm soil sampling depth by citing studies highlighting its sensitivity to anthropogenic and natural disturbances.

Line 144: Provide a detailed rationale for adopting the acid hydrolysis method and discuss its limitations, particularly in differentiating labile from recalcitrant carbon fractions.

Figure 2: Standardize the presentation of Figure 2 by including absolute values, relative percentages, and error bars to improve interpretability. Also, please give a distinction to the subplots.

Figure 4: I am confused about the graph on the right side of Figure 4. How was the data obtained (is it correct)? What does the effect here refer to? Please clarify. Moreover, please expand abbreviations like "Envi" (environmental variables) and "Mic" (microbial indicators) to facilitate interpretation by diverse readerships.

Table 2: Specify the replication number (e.g., n=3) and statistical measures used to ensure transparency in data interpretation.

Discussion: The discussion is the weakest part of this manuscript. A well-structured discussion should be based on this study's findings and combined with previous studies to discuss something new rather than simply listing and piling up the results.

References: Ensure full compliance with journal-specific formatting requirements, standardizing Latin names, capitalization, and citation styles.

General: please conduct a meticulous review for linguistic precision, ensure that terminology aligns with current soil and microbial ecology standards, and refine syntax for enhanced readability and academic rigor.

**Do you want your identity to be public for this peer review?** For information about this choice, including consent withdrawal, please see our Privacy Policy

Reviewer #1: **Yes: ** Rudong Zhao

Reviewer #2: No

---

## [Author Response · Author response to Decision Letter 1]

18 Apr 2025

We sincerely appreciate the editor’s careful review and constructive suggestions regarding the manuscript format and structure. We have thoroughly addressed all the editorial comments.

---

## [Decision Letter · Decision Letter 1]

15 Sep 2025

Dear Dr. Ding,

Thank you for submitting your manuscript to PLOS ONE. After careful consideration, we feel that it has merit but does not fully meet PLOS ONE’s publication criteria as it currently stands. Therefore, we invite you to submit a revised version of the manuscript that addresses the points raised during the review process.

We look forward to receiving your revised manuscript.

Kind regards,

Tiziana Danise, PhD

Academic Editor

PLOS ONE

Journal Requirements:

Reviewers' comments:

Reviewer's Responses to Questions

**Comments to the Author**

Reviewer #2: All comments have been addressed

Reviewer #3: All comments have been addressed

Reviewer #4: All comments have been addressed

Reviewer #5: All comments have been addressed

2. Is the manuscript technically sound, and do the data support the conclusions?

Reviewer #2: Yes

Reviewer #3: Yes

Reviewer #4: Partly

Reviewer #5: Yes

3. Has the statistical analysis been performed appropriately and rigorously?

Reviewer #2: Yes

Reviewer #3: Yes

Reviewer #4: Yes

Reviewer #5: Yes

4. Have the authors made all data underlying the findings in their manuscript fully available?

Reviewer #2: Yes

Reviewer #3: Yes

Reviewer #4: Yes

Reviewer #5: Yes

5. Is the manuscript presented in an intelligible fashion and written in standard English?

Reviewer #2: Yes

Reviewer #3: Yes

Reviewer #4: Yes

Reviewer #5: No

Reviewer #2: The authors have addressed the comments effectively; however, there are still some spelling, formatting, and grammatical errors present. For instance, abbreviations are repeated unnecessarily, and variables are not italicized as required. Specific examples include lines 39, 90, 92, 93, 139, 208, 209, and 214. A thorough review of the manuscript is needed to correct these issues.

Reviewer #3: The manuscript titled "Differences in Soil Carbon Fractions and Microbial Communities and Their Underlying Mechanisms between Assisted Natural Regeneration and Plantation Forests in Subtropical China" presents a comprehensive and well-executed investigation into how different forest management strategies influence soil organic carbon (SOC) fractions and microbial dynamics in a humid subtropical environment. The authors employ a robust experimental design encompassing four forest types, detailed field sampling, precise biochemical analyses, and advanced statistical modeling (including structural equation modeling) to elucidate the biotic and abiotic drivers of SOC stabilization.

The study clearly demonstrates that assisted natural regeneration (ANR) supports SOC preservation by enhancing microbial-mediated carbon retention, increasing litter and fine root inputs, and reducing disturbances typically associated with plantation forestry. The distinction made between labile and recalcitrant carbon fractions, and the finding that labile carbon plays a more critical role in SOC stabilization in these systems, provides meaningful insight into the mechanistic underpinnings of carbon cycling. The SEM approach effectively clarifies the dual regulatory pathways, linking aboveground litter inputs and belowground microbial community structure to SOC dynamics.

The manuscript is well-organized, methodologically sound, and contributes original findings to the fields of soil science, forest ecology, and carbon management. The writing is clear and concise, figures and tables are appropriate and well-interpreted, and the discussion reflects a deep understanding of the topic. Importantly, the study’s conclusions are well-supported by the data and present actionable implications for sustainable forest management and climate change mitigation strategies.

Given the scientific rigor, clarity, and relevance of the findings, I recommend the publication of this manuscript in its current form.

Reviewer #4: Revision no. PONE-D-24-53500R1

“Differences in Soil Carbon Fractions and Microbial Communities and Their Underlying Mechanisms between Assisted Natural Regeneration and Plantation Forests in Subtropical China”

Since Table 1 reflects basic and important information of the standing trees characteristics in different stands. Therefore, I strongly suggest the authors to provide justifications on

1. How ANR forests outperform plantations in terms of belowground input and in comparison, with CCP and CLP.

2. The authors must provide a brief justification in relation with the understory biomass diversity which would enable to understand by the audiences.

3. This justification much be reflected in abstract, as it bears a meaningful insight.

This manuscript is good and covers many aspects of soil. However, I strongly agreed with the reviewer’s viewpoint which lack of sincere justification in the discussion. The revised manuscript provided shows progress.

I hope, a brief improvement of the above given problems could be a well written manuscript.

Reviewer #5: Suggestions for Improvement:

1. ABSTRACT:

• Replace phrases like “remain insufficiently understood” with simpler wording such as “are not yet fully understood”; also break long sentences (e.g., line 24–25) for readability.

• Use abbreviations consistently (e.g., SF) or omit them; ensure species names (Castanopsis carlesii, Cunninghamia lanceolata) are italicized throughout.

• Shorten phrases like “SEM analysis revealed” to “SEM identified”; refine awkward wording such as “no direct influence path” → “no direct pathway of influence”.

• Replace vague terms like “SOC levels” with more specific wording (e.g., “SOC concentrations”).

• The concluding sentence (lines 25–27) reads like part of the discussion; consider moving it to strengthen the conclusion section.

2. INTRODUCTION:

• Some sentences are overly long and could be split for readability (e.g., lines 41–43 and 53–56). Shorter sentences would help highlight key findings more clearly.

• Use abbreviations consistently: SOC, LC, RC, and ANR are well defined, but “secondary natural forest” (line 86) is not abbreviated, unlike elsewhere. Decide whether to introduce abbreviations for all forest types or avoid them.

• Replace repetitive structures such as “remain poorly understood” (line 46) and “remain inadequately understood” (line 75) with varied phrasing to improve flow.

• Line 55: “facilitate improved resources and environmental management” is vague. Consider specifying “facilitate improved soil resource use and ecosystem management”.

3. MATERIALS & METHODS:

• Breaking the longer sentences into shorter steps or using bullet-style formatting for better readability (e.g., lines 150–166, acid hydrolysis method).

• Abbreviations like SF, ANR, CCP, and CLP should be used consistently after introduction.

• In soil and PLFA methods, some units and symbols (e.g., g•cm⁻³, Mg•hm²) could be standardized to SI format for consistency. Check spacing in chemical formulas (e.g., H₂SO₄, NH₄⁺-N).

• Ensure uniform citation of figures and tables (e.g., Fig. 1c vs. Fig 1c; Table 1 is referenced, but caption and format could be improved for clarity and readability).

• Phrasing like “Previous studies have shown…” (line 135) or “In the present study…” (line 149) is slightly repetitive. Consider more varied transitions to improve flow.

4. RESULTS:

• Clarity of Data Presentation: Some descriptions of results (lines 218–225) are overly dense with numbers and percentages. Breaking these into shorter sentences or using parentheses sparingly would improve readability.

• Terminology Consistency: Ensure abbreviations (e.g., NH₄⁺-N, NO₃⁻-N, FRB, LS) are consistently defined once and used uniformly across text, tables, and figures.

• Interpretation Balance: The discussion of plantations emphasizes nutrient and SOC declines but offers little context about variability or possible ecological explanations (e.g., species-specific traits of Castanopsis vs. Cunninghamia). Adding one line of interpretation would strengthen this.

• Figure/Table Referencing: References to Table 2 and Figure 2 are appropriate, but Figure 3 and Figure 4 interpretations (lines 269–303) could briefly emphasize the ecological meaning of the significant pathways rather than focusing mainly on R² values.

5. DISSCUSION:

• The model fit indices (χ², df, RMSEA, CFI, TLI) are reported, but the narrative doesn’t fully explain their ecological significance. Readers unfamiliar with SEM may find it overly technical.

• The explanation of LC and RC dynamics is repeated across multiple paragraphs. This could be condensed to avoid redundancy and to keep focus on novel insights.

6. CONCLUSION:

• The paragraph is breaking it into two shorter paragraphs for better readability and flow.

• Acronyms such as LC, RC, SOC, ANR, GP:GN are frequently used — ensure they are defined at first use within this section for clarity to readers unfamiliar with earlier parts.

• The use of percentages (e.g., “SOC reduced by up to 30.4%”) would benefit from specifying whether this was relative to secondary forests, ANR, or another reference system.

7. SUGGESTIVE NOTE:

I find this manuscript has;

• Scientific Contribution – The focus on SOC fractions (labile vs. recalcitrant) and their microbial underpinnings in subtropical forests is timely and significant. The finding that LC serves as a critical mediator of SOC stability provides fresh insights.

• Methodological Rigor – Use of multi-parameter soil and microbial analyses alongside SEM enhances the robustness of causal inference.

• Novel Insights – The comparison of ANR and plantations highlights both ecological and management implications, especially regarding LC-to-RC conversion pathways.

• Data Presentation – Tables and figures are detailed and well-organized, allowing clear visualization of results.

• Policy Relevance – The work provides applicable recommendations for sustainable forestry and carbon management.

The manuscript makes a valuable contribution to forest ecology, carbon cycle research, and sustainable land management. I recommend minor revisions focusing on clarity, consistency, and contextual framing. Once these improvements are incorporated, the manuscript will be well-prepared for acceptance.

**Do you want your identity to be public for this peer review?** For information about this choice, including consent withdrawal, please see our Privacy Policy

Reviewer #2: No

Reviewer #3: **Yes: ** Tancredo Souza

Reviewer #4: No

Reviewer #5: **Yes: ** Dr. Amina Kanwal

---

## [Author Response · Author response to Decision Letter 2]

30 Sep 2025

Thank you for the opportunity to submit a revised version of our manuscript titled “Differences in Soil Carbon Fractions and Microbial Communities and Their Underlying Mechanisms between Assisted Natural Regeneration and Plantation Forests in Subtropical China” (Manuscript ID: PONE-D-24-53500R1). We sincerely appreciate the constructive comments and suggestions provided by the reviewers and the academic editor.

We have carefully addressed all the points raised during the review process. The main revisions include:

Clarifying methodological details as suggested.

Improving the discussion of results to better highlight their significance.

Ensuring that all conclusions are well-supported by the data.

Minor grammatical and stylistic edits for improved readability.

We believe that the revisions have significantly improved the manuscript and hope that it now meets the publication standards of PLOS ONE.

---

## [Decision Letter · Decision Letter 2]

7 Dec 2025

Differences in Soil Carbon Fractions and Microbial Communities and Their Underlying Mechanisms between Assisted Natural Regeneration and Plantation Forests in Subtropical China

PONE-D-24-53500R2

Dear Dr. Ding,

We’re pleased to inform you that your manuscript has been judged scientifically suitable for publication and will be formally accepted for publication once it meets all outstanding technical requirements.

Kind regards,

Ying Ma, Ph.D.

Academic Editor

PLOS One

Additional Editor Comments (optional):

Reviewers' comments:

Reviewer's Responses to Questions

**Comments to the Author**

Reviewer #2: All comments have been addressed

Reviewer #3: All comments have been addressed

Reviewer #6: All comments have been addressed

2. Is the manuscript technically sound, and do the data support the conclusions?

Reviewer #2: Yes

Reviewer #3: Yes

Reviewer #6: Yes

3. Has the statistical analysis been performed appropriately and rigorously?

Reviewer #2: Yes

Reviewer #3: Yes

Reviewer #6: Yes

4. Have the authors made all data underlying the findings in their manuscript fully available?

Reviewer #2: No

Reviewer #3: Yes

Reviewer #6: Yes

5. Is the manuscript presented in an intelligible fashion and written in standard English?

Reviewer #2: Yes

Reviewer #3: Yes

Reviewer #6: Yes

Reviewer #2: (No Response)

Reviewer #3: I have carefully reviewed the revised manuscript and the authors’ point-by-point responses. The authors have satisfactorily addressed all comments raised during previous review rounds and have substantially improved the clarity, structure, and scientific rigor of the paper.

I recommend the manuscript for publication in its current form.

Reviewer #6: This is a revised manuscript and the authors have made great efforts to address reviewers' comments. I find the research is quite interesting and recommend Accept.

**Do you want your identity to be public for this peer review?** For information about this choice, including consent withdrawal, please see our Privacy Policy

Reviewer #2: No

Reviewer #3: No

Reviewer #6: No

---

## [Editor Report · Acceptance letter]

PONE-D-24-53500R2

PLOS One

Dear Dr. Ding,

I'm pleased to inform you that your manuscript has been deemed suitable for publication in PLOS One. Congratulations! Your manuscript is now being handed over to our production team.

Kind regards,

on behalf of

Dr. Ying Ma

Academic Editor

PLOS One